# Design of a Fully Integrated Inductive Coupling System: A Discrete Approach Towards Sensing Ventricular Pressure

**DOI:** 10.3390/s20051525

**Published:** 2020-03-10

**Authors:** Natiely Hernández Sebastián, Noé Villa Villaseñor, Francisco-Javier Renero-Carrillo, Daniela Díaz Alonso, Wilfrido Calleja Arriaga

**Affiliations:** 1CD-MEMS, National Institute for Astrophysics, optics and Electronics, INAOE, Puebla 72840, Mexico; 2Optics Research Center, CIO A. C., León 37150, Mexico; 3Department of I. T., Electronics and Control, Advanced Technology Center, CIATEQ A.C., San Luis Potosí 78395, Mexico; noe.villa@ciateq.mx; 4Optics department, National institute for Astrophysics, Optics and Electronics (INAOE), Puebla 72840, Mexico; paco@inaoep.mx; 5Microtechnologies Department, Center for Engineering and Industrial Development, CIDESI, Queretaro 76125, Mexico; daniela.diaz@cidesi.edu.mx; 6Science and Technology Biomedics Graduated Program, National Institute for Astrophysics, Optics and Electronics, INAOE, Puebla 72840, Mexico

**Keywords:** integrated coupling system, ventricular pressure sensor, BioMEMS, flexible electronics, wireless power transfer, implantable medical device

## Abstract

In this paper, an alternative strategy for the design of a bidirectional inductive power transfer (IPT) module, intended for the continuous monitoring of cardiac pressure, is presented. This new integrated implantable medical device (IMD) was designed including a precise ventricular pressure sensor, where the available implanting room is restricted to a 1.8 × 1.8 cm^2^ area. This work considers a robust magnetic coupling between an external reading coil and the implantable module: a three-dimensional inductor and a touch mode capacitive pressure sensor (TMCPS) set. In this approach, the coupling modules were modelled as RCL circuits tuned at a 13.56 MHz frequency. The analytical design was validated by means of Comsol Multiphysics, CoventorWare, and ANSYS HFSS software tools. A power transmission efficiency (PTE) of 94% was achieved through a 3.5 cm-thick biological tissue, based on high magnitudes for the inductance (L) and quality factor (Q) components. A specific absorption rate (SAR) of less than 1.6 W/Kg was attained, which suggests that this IPT system can be implemented in a safe way, according to IEEE C95.1 safety guidelines. The set of inductor and capacitor integrated arrays were designed over a very thin polyimide film, where the 3D coil was 18 mm in diameter and approximately 50% reduced in size, considering any conventional counterpart. Finally, this new approach for the IMD was under development using low-cost thin film manufacturing technologies for flexible electronics. Meanwhile, as an alternative test, this novel system was fabricated using a discrete printed circuit board (PCB) approach, where preliminary electromagnetic characterization demonstrates the viability of this bidirectional IPT design.

## 1. Introduction

In recent years, implantable medical devices (IMDs) have undergone rapid progress towards new analysis schemes in modern medical equipment, in which the ability to monitor vital signs and stimulation techniques is improving [1]. Representative examples of IMDs include pacemakers, pressure sensors, spinal cord stimulators, and pumps for drug delivery [2,3]. Currently, the energy of most IMDs is supplied by wiring lines running through the patient’s body or by utilizing integrated batteries. These methods show poor performance, since the use of batteries generates bulky modules and cables increase the possibilities of infection. Additionally, due to the limited life of batteries, replacement surgeries are required, which increases the financial burden for a patient [2,4,5]. A promising alternative to contactless energy supplying for long-term implantable devices is wireless power transfer (WPT) [1,6,7] systems. Inductive power transfer (IPT) is one of the techniques used to deliver power to IMDs wirelessly. In this approach, two mutual inductive coupling devices were linked, following a transformer-type approach, where bidirectional power transmission can be carried out [8]. In such systems, high power transmission efficiency (PTE) and robustness are desirable, usually under restricted room conditions. For these purposes, new inductor devices with both high inductance (L) and quality factor (Q) for advanced design and manufacturing techniques are required. Recently, a large amount of research has been devoted to the improvement of PTE and reducing sensitivity to lateral misalignment [1,2,8,9]. However, several systems are characterized by some complex components and result in large area inductors, which are inadequate for implants [1,2,3,4]. For example, Trigui [10] proposes the use of advanced control circuits to compensate for variations in the resonance frequency of the coils. Although this approach is effective in improving PTE, it was very complicated to implement and calibrate. M. Feenaghty [9] proposed a planar Archimedean coil (PAC), which is implemented for IPT applications; this geometry combines aligned and misaligned conventional coils in a single Archimedean design, resulting in a Q twice higher than that of a traditional coil. Although the system showed an improved PTE, Q is drastically reduced when the secondary coil is misaligned. S. Chun Tang [11] demonstrated that a transmission coil with a larger outer diameter could provide a more extensive power transmission range and lower lateral misalignment sensitivity, but the significant size difference between the coils dramatically reduces the magnetic coupling factor and, therefore, the PTE of the system. In 2004 [12], the development of a new class of implantable devices for aortic aneurysms and heart failure began. The system was named CardioMEMS and consisted of an implantable pressure sensor, an external communication module, and an intravenous system designed to deploy the sensor in the pulmonary artery. The battery-less 3.5 × 30 mm device has a wireless range of about 20 cm. However, it is not considered a continuous monitoring system, since, to obtain a single measurement, the patient must lie on a pillow, which is part of the external communication module, and press a button. In summary, for implantable devices, it is desirable to reduce the size of the coils. To decrease the size of the inductors, some alternatives have been proposed, for example, the use of a higher frequency of operation [13], but it is known that high frequencies increase the absorption of energy in biological tissue, causing adverse health problems and a need for a high operating voltage in the transmitter module [2,10,11,12,13,14,15].

This paper presents a novel approach for the design of an optimized and integrated IMD, combining a low-cost thin-film surface micromachining technique and the use of polymers for flexible electronics. The implantable module was developed containing a 3D inductor-capacitor array, a new integrated L-C approach for the novel performing of an IPT system, offering an optimized continuous monitor of the cardiac pressure for a wide 5–300 torr range. This approach results in a lower area and maximum energy transfer device across the biological tissue, which allows obtaining a precise and low-cost IMD. This proposal maximizes the use of the reduced area and provides robustness and high efficiency of transmission through the biological tissue, allowing an accurate and low-cost implantable sensor device. The experimental approach was performed with a discrete printed circuit board (PCB) circuitry set, emulating the required electromagnetic power transfer and allowing visualization of the projected performance supporting this novel inductive coupling system design.

## 2. Theoretical Inductive Power Transfer Design

The proposed system is considered for implantation into the left ventricle, aiming for continuous sensing of a wide blood pressure range of 5–300 torr [16]. This application imposes an important floor limitation for the sensor area, 1.8 × 1.8 cm^2^, which demands new design features for the telemetric design. The design considers that the external coil is located on the outer surface of the body and the implantable coil is placed inside the left ventricle at a depth of 3.5 cm. This distance corresponds to the combined thickness of the skin, fat, and muscle, separating both coils. The two RCL circuits are tuned to a resonance frequency of 13.56 MHz to attain the best PTE. According to the ISO 14,117 standard, this frequency does not induce tissue damage by radiation and heating, nor cause interference with other implantable medical devices. Table 1 shows the constitutive parameters of the biological tissue used as the magnetic core.

Figure 1 shows the equivalent electrical model for the wireless ventricular pressure sensor array. The system consisted of two passive RCL circuits designed to transfer power through biological tissue. The mechanical action was accomplished by a pressure sensor array and the inductive power action was performed by a novel structured coil. According to anatomy room restriction into the left ventricle, the design procedure was upgraded to attain an effective telemetry device. Details of this system design are discussed hereafter.

### 2.1. Mechanical Sensing Action

In this design, the pressure sensitive element $C_{s}$ was a touch mode capacitive pressure sensor (TMCPS). Initially, the TMCPS sensor was modeled as a parallel-plates variable capacitor, where the upper plate or diaphragm was touching and gradually positioned over the isolated lower plate in response to some pressure increase into the ventricle. This mechanical action as a function of pressure gives place to a corresponding change in capacitance (C) [16]. In this model, the value of C at 0 pressure is equal to:(1)$$C_{p=0}\;=\;\frac{\varepsilon_{o}\varepsilon_{r}A}{t_{d}},\;$$
where $\varepsilon_{0}$ is the permittivity of the free space, $\varepsilon_{r}$ is the relative permittivity of the dielectric, A is the contact area between the plates, and $t_{d}$ is the total thickness (separation) of the dielectric. Since $\varepsilon_{0}$ and $t_{d}$ are constants, the capacitance only depends on the contact area.

### 2.2. Inductive Power Calculation

The TMCPS sensor device was electrically connected to a multilevel planar coil. This circuit formed a parallel RCL array with an operating frequency f that was pressure dependent [16,17,18,19,20], according to:(2)$$f\;=\;\frac{1}{2\pi }\left(\frac{1}{\sqrt{{\mathrm{LC}}_{S}}}\right),$$
where L is the inductance of the coil and C is the total capacitance of the TMCPS array. The magnetic coupling between the implantable coil $\left(L_{s}\right)$ and the reading coil $\left(L_{r}\right)$ can be modeled as a two-port network, according to the arrangement of Figure 1. By applying the Kirchhoff voltage law, and by considering that s = jω = j2πf, it yields:(3)$$V_{r}{\;=\;\mathrm{sL}}_{r}I_{r}{\;+\;\mathrm{sMI}}_{s},$$
(4)$$V_{s}{\;=\;\mathrm{sMI}}_{r}{\;+\;\mathrm{sL}}_{s}I_{s}.\;$$

In Equations (3)–(4), $V_{r}$ and $I_{r}$ are the voltage and current at the reader coil, respectively; $V_{s}$ and $I_{s}$ are the voltage and current in the implantable coil, respectively; M is the mutual inductance between the coils; and $Z_{s}$ is the equivalent impedance of the implantable set. The secondary current $I_{s}$ and the secondary voltage $V_{s}$ can be calculated from Equations (1)–(4), as follows:(5)$$I_{s}{\;=\;I}_{r}\frac{-\mathrm{sM}}{R_{s}{\;+\;\mathrm{sL}}_{s}{\;+\;Z}_{s}}.$$

By substituting Equation (5) into Equation (2) the voltage transfer function can be derived as:(6)$$\frac{V_{s}}{V_{r}}\;=\;\frac{{\mathrm{sMZ}}_{2}}{\omega^{2}\left(M^{2}{-L}_{r}L_{s}\right)+s\left(L_{r}R_{s}{+L}_{2}R_{1}{+L}_{1}Z_{s}\right)+\left(R_{1}R_{2}\right)+\left(R_{1}Z_{2}\right)}.$$

A usual parameter is the magnetic coupling factor k between two coils. In this sense, k relates the mutual inductance for Equation (6). Physically, k is equal to the fraction of magnetic flux generated by the $L_{r}$ coil, which flows through the secondary $L_{s}$ coil. If the external radius of the reading coil is denoted as $R_{\mathrm{out}.r}$, then the external radius of the internal coil is denoted as $R_{\mathrm{out}.s}$, and if the distance between the two coils is denoted by X, then, for an implantable system, in which, generally, $R_{\mathrm{out}.r}>R_{\mathrm{out}.s}$, k can be approximated by:(7)$$k\;=\;\frac{R_{\mathrm{out}.r}^{2}{\;\times \;R}_{\mathrm{out}.s}^{2}}{\sqrt{R_{\mathrm{out}.r}R_{\mathrm{out}.s}{\left(X^{2\;}{+\;R}_{\mathrm{out}.r}\right)}^{3}}}.\;$$

The quality factor for each coil is given by:(8)$$Q\;=\;\frac{1}{R}\sqrt{\frac{L}{C}}.$$

Relating the coupling factor with the mutual inductance between the coils and replacing Equations (2) and (8) in Equation (6), the equivalent impedance $Z_{r}$ for the link can be approximated as follows:(9)$$Z_{r}\;=\;\frac{V_{r}}{I_{r}}{\;=\;j2\pi \mathrm{fL}}_{r}\left[{1\;+\;k}^{2}\frac{{\left({f/f}_{s}\right)}^{2}}{1\;-\;{\left({f/f}_{s}\right)}^{2\;}+\;\left(1/Q\right)j\left({f/f}_{s}\right)}\right].\;$$

This equation relates the resonance frequency $f_{s}$ of the implantable device with an electrical variable measurable from the reading device. That is, the pressure changes at the biological medium (see Equation (1)) can be related to the changes in the equivalent impedance $Z_{r}$ as a function of $f_{s}$. Therefore, pressure variations in the left ventricle can be measured indirectly in the reading device.

### 2.3. Electromagnetic Flux Coupling

In order to perform indirect measurements and transfer power to the implantable set, a variation of the Maxwell–Wien bridge circuit was used, as shown in Figure 2. The circuit in Figure 2 allows measuring an unknown inductance in terms of known resistance and capacitance parameters. According to Figure 2, $R_{1}$ and $L_{1}$ have fixed and known magnitudes and must be the same as those for the reading device. The magnitudes of $R_{2}$, $R_{3}$, and $C_{r}$ were variable, and their final was those that allowed the equilibrium at the bridge circuit.

In the case of absent signal from the implantable set, the bridge circuit could be balanced, and the output voltage across points B–C was 0. Once the coils $L_{r}$ and $L_{s}$ were magnetically coupled, the coupling factor k was different from 0, and $Z_{r}$ changed accordingly. If the change in $Z_{r}$ is significant, then it will produce a proportional output voltage across points B–C.

Once the sensor set was implanted, the distance between the coils remained constant, and the value of k did not change accordingly. However, if the ventricular pressure varied, the resonance frequency $f_{s}$ at the implantable set will change as a function of $C_{s}$. The change in $f_{s}$ was detected as a variation in $Z_{r}$, and, in turn, a change in the voltage between the B–C nodes at the reader device was registered.

### 2.4. Electromagnetic Flux Calculation

The electromagnetic flux coupling was designed by considering the properties of near field electromagnetic alignment (Fresnel), based on circular geometry coils and by considering a restricted physical floor of 18 mm × 18 mm for the implantable device. The design details for each coupling module are described in the following subsections.

#### 2.4.1. RCL Implantable Set

The implantable assembly consisted of a three-dimensional planar coil, a variable capacitor (pressure sensor), and a resistive load. The three parallel-connected elements formed a passive RCL tank circuit. This subsystem allowed wireless communication with the external RCL circuit. In this design, the use of a two-level planar coil maximized the use of space and enabled high values of inductance and quality factor, which, in turn, improved the PTE and the robustness of the system.

The self-inductance of a single-layer planar coil can be calculated from the following equation [21]:(10)$$L\;≅\;\frac{\mu_{0}N^{2}D_{\mathrm{avg}}C_{1}}{2}\left[\ln\left(\frac{C_{2}}{F}\right){\;+\;C}_{3}{F\;+\;C}_{4}F^{2}\right],$$
where $\mu_{0}$ is the permeability of the free space, $N=\left(R_{\mathrm{out}}-R_{\mathrm{in}}\right)/\left(w+s\right)$ is the number of turns of the inductor, w and s are the width and separation turns of the coil, $D_{\mathrm{avg}}=\left(D_{\mathrm{out}}+D_{\mathrm{in}}\right)/2$ is the average diameter of the coil, $F=\left(D_{\mathrm{out}}-D_{\mathrm{in}}\right)/\left(D_{\mathrm{out}}+D_{\mathrm{in}}\right)$ is a parameter known as the fill factor and coefficients, and $C_{1}$ to $C_{4}$ are constants determined by the geometry of the coil. For a circular planar coil, $C_{1}=1$,${\;C}_{2}=2.46$, $C_{3}=0$ y $C_{4}=0.2$.

The self-inductance of the two-level coil could not be calculated using Equation (10), because this kind of coil generated a mutual inductance between its turns, or rather, the self-inductance is calculated as [15,22,23]:(11)$${L\;=\;L}_{1}{\;+\;L}_{2}\;\pm \;2M,\;$$
where $L_{1}$ and $L_{2}$ are the self-inductance of each level, calculated by using Equation (10), and $M\;=\;k{\left(L_{1}{\;+\;L}_{2}\right)}^{1/2}$ is the mutual inductance between levels. From Equations (10) and (11), if the effective mutual coupling between the coils is positive, L can be increased by augmenting the number of turns N. This concept was used to achieve the design parameters required for the two-level inductor, trying to obtain the best quality factor. In the proposed design, the width of the metallic line $w\;=\;\left[\left(R_{\mathrm{out}}{\;-\;R}_{\mathrm{in}}\right)/N\right]\;-\;s$ is a key parameter. $R_{\mathrm{out}}$ is the outer radius, $R_{\mathrm{in}}$ is the internal radius, N is the number of turns, and s is the spacing between the metal lines. In order to obtain the higher magnetic coupling, it is mandatory that w ≤ s and the internal length l cover at least 1/4 of the wavelength λ associated with the resonance frequency of the RCL circuit. The last parameter is determined as:(12)$$\lambda \;=\;\frac{V}{f},\;$$
where V = c/n is the speed of propagation for the waves at the transmission medium. For the proposed IPT system, f = 13.56 MHz, and the signal propagates through the combination of three types of biological tissue: skin, fat, and muscle. For the implantable coil, a wavelength λ = 5.7 m was obtained, and the coil had at least a length of 1.4 m. Because of the restricted floor for this design, the width of the metal line w was crucial to cover the required length. For a circular planar coil, where w = s, the length of the coil could be determined from the analysis of an Archimedean spiral, defined as [23]:(13)$${l\;=\;R}_{\mathrm{in}}2\pi N\;+\;\frac{s}{4\pi }{\left(2\pi N\right)}^{2},\;$$

The two-level metallic spiral was symmetrically arranged, avoiding overlapping in order to obtain the highest efficiency for the coil. Each level was made up of 28 turns of 160 μm width, $R_{\mathrm{in}}\;=\;1\;\mathrm{mm}$, and an external radius of 9 mm. The two spirals, interconnected in a series, structured with aluminum (Al) and covered with a polyimide (PI) film, were designed considering a thin-film monolithic manufacturing approach. This technology allows the definition of capacitive structures (the TMCPS) and inductive structures in a single flexible ergonomic substrate, avoiding the use of hybrid-type electrical wiring. Figure 3 shows the design features for the proposed two-level inductor.

This inductor approach allows high magnitudes for L and Q, considering reduced floor and minimizing parasitic capacitances between both metallic levels, because the spirals were traced, avoiding overlapping. As a result, the PTE and robustness of the system were improved.

#### 2.4.2. RCL External Device

The external RCL device was located outside the human body and was designed under wide flexibility in terms of materials and structural dimensions. The design consisted of a single-layer coil, a variable capacitor $C_{r}$, and a resistive load $\left(R_{r}\right)$. For the external coil $L_{r}$, the design was made by considering two concepts. The first was to determine the size of the planar coil as a function of the radiation distance (X) and the tissue losses. We can then approximate the external radius by using [24]:(14)$$R_{\mathrm{out}.r}\leq X\sqrt{2,}\;$$
where $R_{\mathrm{out}.r}$ is the external radius of the coil and X is the radiation distance. Since equation (14) allows obtaining only a range of possible values for $R_{\mathrm{out}.r}$, the optimal value was determined from the complementary analysis for the magnetic field strength H, according to the following equation:(15)$$H\;=\;\frac{I_{r}{\times N\times R}_{\mathrm{out}.r}^{2}}{2\sqrt{{\left(R_{\mathrm{out}.r}^{2}{+X}^{2}\right)}^{3}}}.\;$$

If the distance separation between both coils is kept constant and $R_{\mathrm{out}.r}$ is varied under the assumption of a constant electrical current, then the intensity of H reaches its highest magnitude at a single ratio for X and $R_{\mathrm{out}.r}$ [25]. Therefore, for each radiation distance, there was an optimum external radius for the reading coil. Figure 4 shows the H strength as a function of the external radius for three different radiation distances.

In Figure 4, the modeled system considers that the electric current was I = 1 and the number of turns was N = 27. If the external radius of the reading coil was too large, then the intensity of the field was too low, even at a 0 distance. On the other hand, if the radius of the reading coil was small, then the H intensity amounted to a ratio of $x^{3}$. For a radiation distance of X = 3.5 cm, an optimum radius of 4 cm was determined.

The second concept considered the geometrical matching between both coils, $L_{r}$ and $L_{s}$, in order to obtain the highest coupling factor. Accordingly, the resonance frequency $f_{r}$, the mutual inductance M, and the internal radius of the external coil were determined following the implantable coil parameters. Under these considerations, the single-layer external coil was projected over a FR-4 printed circuit board as substrate material. The copper coil 8cm × 8cm area was structured by 27 turns with a 700 μm-wide and 35 μm-thick copper stripe. The design included several ports for the electrical connection of some discrete components, a power supply, and a signal register. Figure 5 shows the design characteristics of the external coil.

### 2.5. Results

The mutual inductance between both coils can be approximated by using the following equation:(16)$$M\;=\;2\;\times \;k\;\times \;\sqrt{L_{r}{\;+\;L}_{s}},\;$$
where $L_{r}$ and $L_{s}$ are the self-inductance magnitudes for the external and internal coils, respectively, and k is the coupling factor calculated as a function of the geometrical dimensions of the coils (see Equation (7)). To validate M and k factors, Equations (7) and (16) were modeled in OriginPro software, and the result is shown in Figure 6.

For the designed system, in which X = 3.5 cm, a coupling factor of 0.16 and a mutual inductance of 6.48 μH were obtained. The power transmission efficiency (PTE) of the system was calculated as follows:(17)$$\eta \;=\;\frac{k^{2}Q_{r}Q_{s}^{3}R_{s}R_{\mathrm{load}}}{\left(k^{2}Q_{r}Q_{s}^{3}R_{s}R_{\mathrm{load}}{\;+\;k}^{2}Q_{r}Q_{s}R_{\mathrm{load}}^{2}{\;+\;2Q}_{S}^{2}R_{S}R_{\mathrm{load}}{\;+\;R}_{\mathrm{load}}^{2}\right)},\;$$
where $R_{\mathrm{load}}\;\geq {\;2\omega L}_{s}$ is the load resistance, $R_{s}\;=\;\left(\rho l\right)/\omega \delta \left({1-e}^{-h/\delta }\right)$ is the resistance of the implantable RCL circuit, and $Q_{s}$ is the quality factor of the $L_{s}$ coil. For this system, it was analytically calculated that $R_{\mathrm{load}}\;=\;1.5\;KΩ$, Rs = 171 Ω and $Q_{s}\;=\;11.5$. By considering these parameters, the system showed a maximum PTE of 92% and a minimum of 74.5%. This PTE fluctuation resulted from the controlled transition into the TMCPS capacitance as a function of the surrounding biological pressure. In other words, the higher the pressure the higher the capacitance; Q and PTE then decreased, and vice versa (see Equations (8) and (17)). 

As a general conclusion, Figure 7 shows the relationship between the quality factor Q, the power transmission efficiency PTE, and the capacitance of the TMCPS. Table 2 shows the main parameters for both internal and external coils. 

## 3. Electromagnetic Flux Simulation

The analytical calculation for the bidirectional IPT system was validated by simulation in Comsol Multiphysics and ANSYS HFSS software. This scheme considered the following modules: an implantable two-level planar circular coil, structured with aluminum films over a polyimide flexible substrate, and an external planar circular coil, structured with a copper film over a PCB plate. The frequency was fixed at 13.56 MHz. A composed biological tissue (skin, fat, and muscle) was considered as the magnetic core for this simplified but effective IPT system (see Table 1).

At first, the simulation of each module was carried out separately to determine the best configuration for the TMCPS and the two-level coil array. After that, the simulation of the complete system was performed. In Figure 8, the main simulation results are shown. A good correspondence can be observed between the simulation results and the electromagnetic parameters can be analytically predicted. 

A variation of less than 2% is observed. In the case of the implantable coil, the results of the simulation show that the proposed coil, composed by two levels, allowed obtaining a quality factor twice as high as that corresponding for the single-level coil. This was due to the calculated 3D structure allowing a combined high inductance and a low electrical resistance. In this way, the proposed approach maximized the performance of the implantable coil and consequently improved the performance of the whole system.

Figure 9 shows a simulated graph of the magnetic flux intensity at a frequency of 13.56 MHz. The simulation included a normal plane, projected from the external coil, located at the center of both coils, which allowed visualizing the strength of the magnetic flux through the biological tissue. The normal plane was projected at a 5 cm distance, where the implanted coil was at 3.5 cm distance (0.5 cm skin, 1 cm fat, and 2 cm muscle). According to the simulation, the maximum intensity of magnetic flux is irradiated at a 4.5 cm distance. However, it was estimated that the designed system covered a radiation distance of at least 6 cm. For the range of 3.5 cm, where the internal coil was located, a magnetic field strength of approximately 2 A/m was obtained.

However, the internal device operated at a 4.5 MHz to 13.56 MHz range. Capacitance changes modulated the frequency of operation; hence the magnetic field distribution was changed due to the equivalent impedance changes reflected at the external coil. Figure 10 shows the simulated variations in the distribution of the magnetic field intensity at the frequencies of 4.5 MHz, 7 MHz, 9.5 MHz, and 12 MHz.

As can be seen, the radiation intensity decreased as a function of frequency. However, the maximum magnetic field strength was preserved at a distance of 3.5 cm, which guarantees the maximum power transfer with the internal coil. Another parameter that was influenced by the frequency changes was the PTE. Figure 11 shows the performance of the PTE system according to the operating frequency. The analysis showed a PTE between a 72.8% and 94.1% range. As was evidenced, the lower the frequency, the lower the PTE; however, this regime was well suited for this type of application.

An essential parameter in wireless power transfer systems is the evaluation of the specific absorption rate (SAR), defined as the measure of power absorbed by the biological tissue. Excessive SAR can lead to an increase in tissue temperature leading to some level of damage. According to the IEEE standard, the SAR should not exceed 2 W/kg for a general safety condition [25].

A simulation routine was proposed to quantify the average SAR, which consisted of a biological tissue filling the gap coils. For our conditions, Figure 12 shows the distribution and the SAR level at a frequency of 13.56 MHz.

In the simulation results, an average SAR value of $4{.8\;\times \;10}^{-1}$ W/Kg was obtained, which was below the standard value (2 W/Kg). Therefore, the proposed IPT system as a means of feeding and transmitting data for the ventricular sensor did not cause tissue damage by the radiation level. Additionally, it was observed that this low SAR was distributed mainly in the skin and fat tissue. Therefore, the power absorption at the muscle (most of the biological tissue) should be discarded.

## 4. Experimental Coupling Evaluation

The implantable set was fabricated and validated for a full system fabrication, and in order to experimentally approach the performance of this novel implantable IPT module, a discrete version was assembled, following the parameters design described in Table 2 over an adapted version of a printed circuit board (PCB). The reading module composed by the Maxwell–Wien bridge circuit and the external coil were also fabricated over a PCB, according to the layout design (Figure 5 and Table 2). 

### 4.1. Experimental Methodology

#### 4.1.1. Implantable Coil Over PCB

The two-level implantable coil was designed using a thin film monolithic manufacturing approach. However, the characteristics of this composed coil were adapted for a standard PCB manufacturing process. The modified layout maintained the electrical and magnetic parameters of the original design (see Table 2) and considered the restricted floor dimensions at the PCB in order to obtain similar spatial performance between both implanted and reading coils. This meant that the internal coil was printed over the same restricted area, modifying the geometry but maintaining the electromagnetic parameters design. The adapted PCB parameters are shown in Table 3.

#### 4.1.2. The External Module

The bridge circuit was constructed by following the design of Figure 2, by considering that $R_{r}{\;=\;R}_{1}{,\;R}_{2}{\;=\;R}_{3}$, and $L_{1}{\;=\;L}_{r}$. Table 4 shows the magnitudes of the electronic components for the bridge circuit.

The Figure 13 shows the bridge circuit, the internal and external coils, and the test configuration for the discrete IPT system. The discrete coils were characterized using a PM6303A automatic impedance analyzer. The electromagnetic parameters showed a maximum variation of 2% in comparison with the analytical results presented in Table 2 and Table 3. Therefore, we have confidence in the discrete components fabricated over the PCB version. 

For the electromagnetic test, a waveform generator powered the bridge circuit and an oscilloscope measured the electrical output signal. In order to avoid degradation of the testing signals, SMA connectors and low-noise cables were used. A sine wave signal, at a frequency of 13.56 MHz and 5-volt peak-to-peak, powered the bridge circuit. The frequency was fixed according to the parameters design.

### 4.2. Experimental Results

#### 4.2.1. Inductive Coupling

The first experiment was conducted for inductive coupling validation purposes. In this geometry array, the internal coil was parallel and facing the coil plate of the Maxwell–Wien bridge circuit. The induced voltage measurements $V_{i}$ were obtained for different separation distances between the coils, starting at 2 mm and ending at 11.5 cm, in steps of 0.5 cm. For comparison purposes, we chose air and a synthetic biological tissue as the magnetic core. Regarding the synthetic tissue, a phantom model composed of pigskin and Agar was used, approaching in thickness and composition to the human tissue. Figure 14 shows the variations of $V_{i}$, measured at the internal coil as a function of distance separation (the magnetic core). For the purpose of performing the synthetic tissue experiments, the maximum available tissue thickness was 3.5 cm, and the rest was (synthetic 3.5 cm + air) step by step until the full 11.5 cm distance for the magnetic core was accomplished. In order to realize this experimental set, the synthetic film was attached to the internal coil and the external coil was the skipping element.

According to the experimental results, $V_{i}$ showed an excellent magnitude around the theoretical magnetic core distance, where this signal appeared gradual and linear around this point. If this distance was increased up to 7 cm, then the obtained signal monotonically tended to disappear. For separation distances greater than 8 cm, the magnetic coupling was practically 0. By comparing the magnitudes for magnetic coupling through the air and synthetic tissue, it was observed that $V_{i}$ was slightly higher through the biological tissue. This behavior was mainly due to some confinement of the magnetic field through this solid core. Across the air, magnetic lines were transmitted in all directions, while in the synthetic biological tissue, the magnetic lines were distributed towards the implantable coil (see Figure 9). In summary, for the radiation distance of 3.5 cm, a $V_{i}$ of 130 mV and 200 mV was obtained through the air and biological tissue, respectively. According to the above, it was concluded that the magnetic coupling between coils was better when the transmission medium was biological tissue.

#### 4.2.2. Capacitance Factor

As already mentioned, changes in the TMCPS array ($C_{s}$) changed the resonance frequency ($f_{s}$) of the implantable device. Such changes were reflected at the output voltage ($V_{\mathrm{out}}$), measured at terminals B and C of the Maxwell–Wien bridge circuit. According to the above, characterization of $V_{\mathrm{out}}$ was made applying changes in capacitance and frequency f into the implantable set. This characterization was carried out for different separation distances between the coils, and, additionally, the synthetic tissue and air were used as the magnetic core. Figure 15 shows the experimental $V_{\mathrm{out}}$ as a function of $f_{s}$ and $C_{s}$. The overall experimental results were very similar in both magnetic core alternatives; therefore, only the results for the magnetic coupling through biological tissue are presented. 

#### 4.2.3. Misalignment Response

Because the pressure sensor was designed for continuous monitoring of ventricular pressure, it was necessary to analyze the IPT performance under misalignment conditions. This was because the left ventricle conducted a turbulent and continuous blood flow. Therefore, some misalignment between the transverse axes of both coils easily occurred. Characterization of possible magnetically decoupling between the coils was carried out by squeezing the transverse axis in distances of 0 cm to 8.5 cm. For each step of misalignment (Z), $V_{i}$ was measured using air and synthetic tissue as the transmission media. Figure 16 shows the obtained results. As expected, the magnitude of this transverse misalignment between the coils affected the performance of the IPT system. For misalignments lower than 3.5 cm, the impact was very low for the system response, since a significate $V_{i}$ was obtained to feed the capacitive array and generate a significate electromagnetic response. However, for a 4 cm and higher misalignment, the lateral displacement played an important role, since the reduced $V_{i}$ signal could be confused with the reference signal of the measuring equipment, which was approximately 20 mV. Therefore, it was concluded that this discrete-like IPT system can transmit-receiving data through two coils under a transverse misalignment no greater than 3.5 cm.

#### 4.2.4. Combined Electromagnetic Core

Regarding the experiments for inductive coupling validation (Section 4.2.1), an additional experimental set was performed, consisting of analyzing the electromagnetic coupling through additional materials. The experimental array was similar to that described in Section 4.2.1, and the results are presented in Figure 17. As shown in this figure, the measurement of the induced voltage V_i_ was performed as follows. The B graph represents both coils separated by a container with Hartmann solution as the magnetic core, realized in steps of 0.5 cm. The A graph represents the synthetic tissue, increasing from 2 mm to 3.5 cm thickness, and then the rest of distance separation was increased step by step across air. The A+B graph represents the synthetic tissue, from 2 mm to 3.5 cm thickness, and then the rest of distance was stepped with the Hartmann solution. The A+B+Synthetic skin graph represents the 3.5 cm synthetic tissue added to another polymer named synthetic skin with 2 mm thickness, and then the full distance was filled with Hartmann solution. As can be seen, the best transmitting signal corresponds to the A graph and the worst transmitting signal condition is across the full Hartmann solution array. This is expected because of the aqueous ionic composition. It is worthwhile mentioning the transmitted strength of the A signal, which is very near to that of the B graph, clearly surpassed the required 3.5 cm of biological tissue for effective coupling. The other two graphs are intermediate and very similar because the magnetic core contained a mixed synthetic material and the aqueous ionic liquid. The inductive coupling across the full-ionic solution, as the worst transmitting condition, shows excellent electromagnetic signal transmission. With this experimental work, we are then able to demonstrate the feasibility of our proposed IPT module.

## 5. Discussion 

This work is related with the field of continuous and ambulatory systems for cardiac pressure monitoring, which today is lacking a well-developed tool for medical assistance. A full-integrated inductive coupling system, intended for the continuous monitoring of cardiac pressure, including a wide range touch-mode capacitive pressure sensor, adapted to 1.8 × 1.8 cm^2^ implanting area, is discussed, attending the proposal for implanting into the left ventricle. Several details for biocompatibility are considered and fulfill regulations, according to IEEE C95.1 guidelines. This novel system includes the following: an integrated internal capacitive sensor and inductor array and an external reading module.

With the purpose of testing feasibility, a discrete approach was developed over a standard PCB substrate, based on a precise calculation emulating the original integrated design. Some key steps were as follows. First, the two-level implantable coil, designed following a thin film monolithic manufacturing approach, was modified, preserving the electrical and magnetic parameters (see Table 2 and Table 3) and considering the restricted floor dimensions in order to obtain similar spatial performance between both the implanted and external reading coils. Second, the reading coil was fabricated according to the original parameters design presented in Table 2. Third, in order to complete the discrete approach, the bridge circuit was constructed by following the design in Figure 2 and includes $R_{r}{\;=\;R}_{1},{\;R}_{2}{\;=\;R}_{3}$, and $L_{1}{\;=\;L}_{r}$, which details regarding magnitudes for the electronic components, as presented in the Table 3. This overall discrete approach shows the following significate electromagnetic results. 

At first, verifying the inductive coupling communication, the internal coil was parallel and faced the coil plate of the Maxwell–Wien bridge circuit. The induced voltage measurements $V_{i}$ were then obtained for different separation distances between both coils, ending successfully at a 5 cm distance separation. For comparison purposes, we chose air, a synthetic biological tissue, and the Hartmann solution as the magnetic core. According to the experimental results, $V_{i}$ shows an excellent magnitude around the theoretical distance for the radiation distance of 3.5 cm, and a $V_{i}$ of 130 mV and 200 mV was obtained through the air and biological tissue, respectively, and the results agree fairly with the Hartmann solution. If this distance increased up to 7 cm, then the obtained signal monotonically tended to disappear. By comparing the magnitudes for magnetic coupling through the air and synthetic tissue, it was observed that $V_{i}$ was slightly higher through biological tissue. This behavior was mainly due to some confinement of the magnetic field through this solid core. 

According the system design, changes into the TMCPS ($C_{s}$) pressure sensor changed the resonance frequency ($f_{s}$) of the implantable device, and such changes were reflected at the output voltage ($V_{\mathrm{out}}$), measured at terminals of the Maxwell–Wien bridge circuit. In concordance, characterization of $V_{\mathrm{out}}$ was made, applying changes in capacitance Cs and frequency f  into the discrete implantable set. This characterization was carried out for the same range of separation distances between the coils. For 3.5 cm of separation distance, it showed a $V_{\mathrm{out}}$ of 500–535 mV range for a corresponding 13.56–4.4 MHz frequency range, also under the corresponding capacitance of a 4–30 pF range. 

Because the integrated inductive coupling system was designed for continuous monitoring of ventricular pressure, it was necessary to analyze the IPT performance under misalignment conditions. This was because the left ventricle conducted a turbulent and continuous blood flow. The analysis of the magnetic decoupling between the coils was then carried out by displacing the transverse axes of one coil against the fixed one in distances of 0 cm to 8.5 cm. For misalignments lower than 3.5 cm, the impact was very low for the system response, since a significate V_i_ was obtained to feed the capacitive array and generate a good enough electromagnetic response. However, for 4 cm and higher misalignments, the lateral displacement played an important role, since the reduced V_i_ signal could be confused with the reference signal of the measuring equipment, which was approximately 20 mV. Therefore, we can conclude that this discrete-like IPT system can transmit receiving data through two coils under a transverse misalignment no greater than 3.5 cm, which is a high window for possible misaligning conditions.

This experimental result is considered as promissory, because the only current commercial system, named CardioMEMS, offers the monitoring of a 0–80 mmHg pressure range with some limited conditions for practical functionality, while our system proposes a continuous and full monitoring 5–300 mmHg blood pressure range, which opens wide possibilities for close monitoring of several cardiac diseases. 

## 6. Conclusions

A fully integrated inductive coupling system, intended for the continuous monitoring of a 0–300 mmHg blood pressure range, including a touch-mode capacitive pressure sensor, adapted to 1.8 × 1.8 cm^2^ internal area, was discussed, attending the proposal for implanting into the left ventricle. Several details for biocompatibility were considered and fulfilled regulations according to IEEE C95.1 guidelines.

After the electromagnetic characterization of this discrete-like inductive coupling system, which closely emulated the overall parameters design, we can conclude and generate excellent expectations on the performance of the integrated version, offering a PTE of 94% through a 3.5 cm-thick biological tissue, based on high magnitudes for the integrated L and Q components. Additionally, a specific absorption rate (SAR) less than 1.6 W/Kg could be attained, which suggests that this IPT system can be implanted in a safe way, satisfying the IEEE C95.1 safety guidelines. According the overall experimental results for this new implantable IPT system, for sensing blood pressure on a continuous basis, the expectative is to overcome the blood pressure range sensing offered by other current commercial systems.

## Figures and Tables

**Figure 1 sensors-20-01525-f001:**
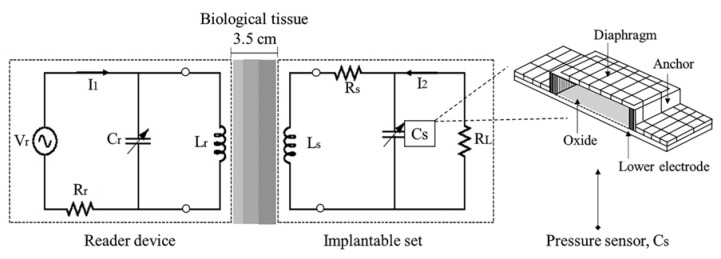
Electrical model for the inductive power transfer system.

**Figure 2 sensors-20-01525-f002:**
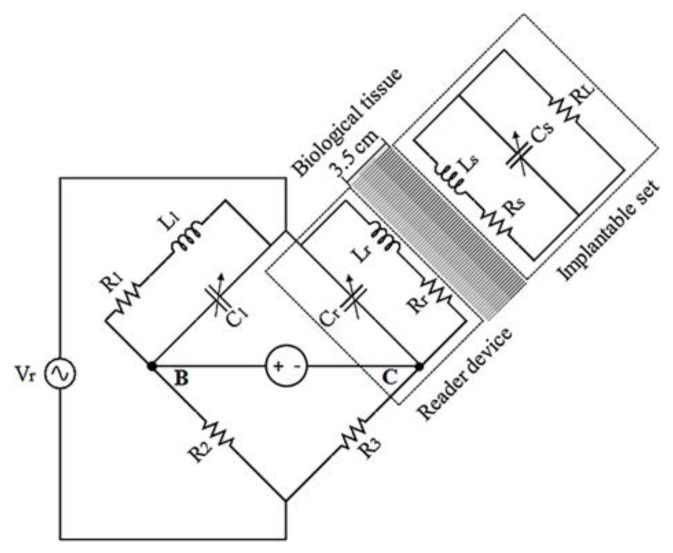
Inductive-coupling array (Maxwell–Wien bridge circuit) to perform transfer power and indirect ventricular pressure measurements.

**Figure 3 sensors-20-01525-f003:**
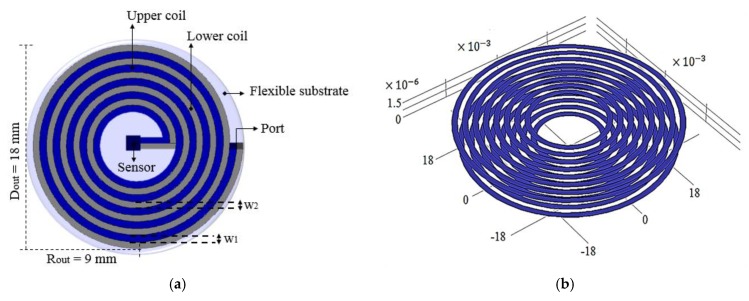
(**a**) Top view and (**b**) 3D view of the designed two-level inductor array.

**Figure 4 sensors-20-01525-f004:**
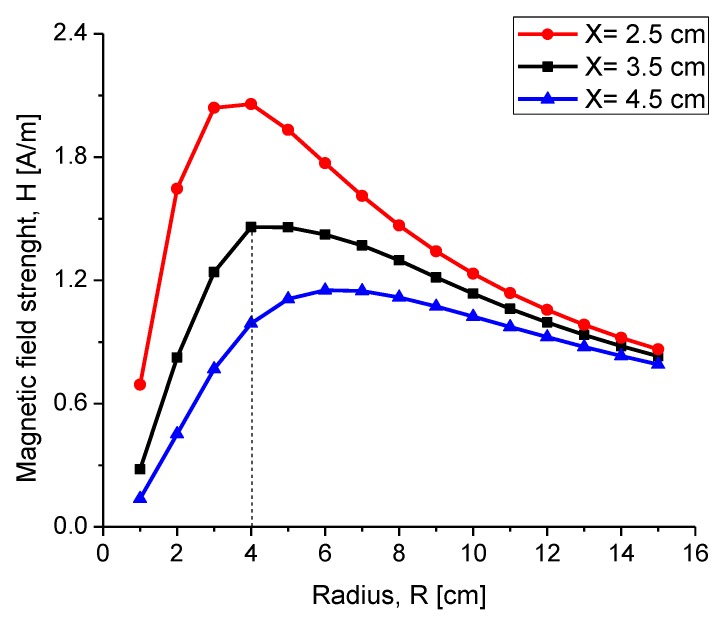
The intensity of field H for the external coil as a function of external radius $R_{\mathrm{out}.r}$ considering three radiation distances X.

**Figure 5 sensors-20-01525-f005:**
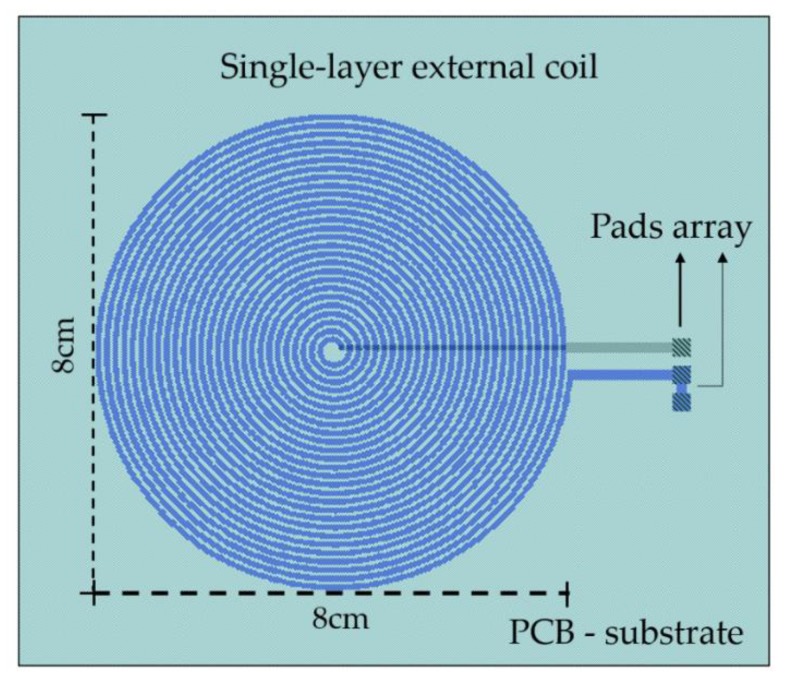
External coil design features.

**Figure 6 sensors-20-01525-f006:**
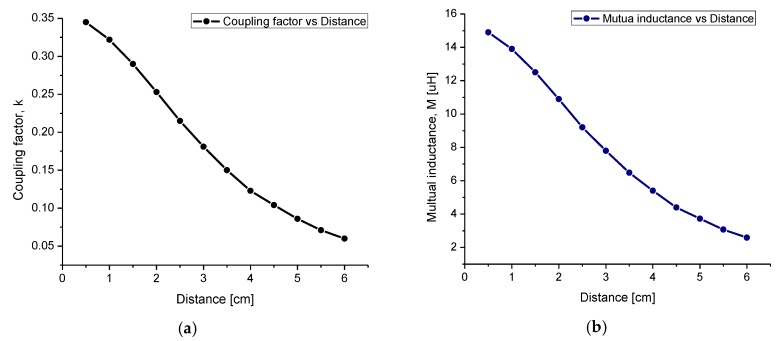
The relationship of both parameters: (**a**) the coupling factor and (**b**) the mutual inductance as a function of distance separation between the two coils.

**Figure 7 sensors-20-01525-f007:**
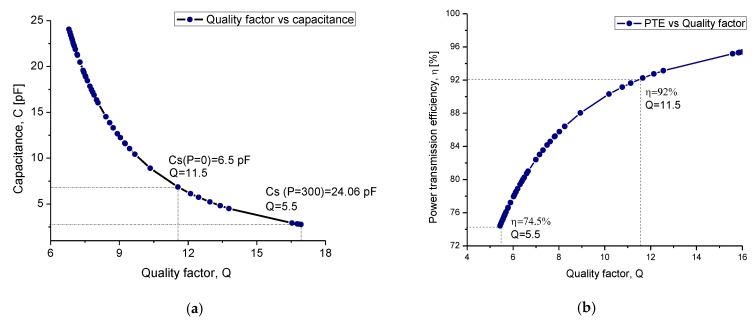
(**a**) The quality factor as a function of the capacitance variations in the touch mode capacitive pressure sensor (TMCPS). (**b**) The relationship between the internal quality factor and the power transmission efficiency (PTE) system.

**Figure 8 sensors-20-01525-f008:**
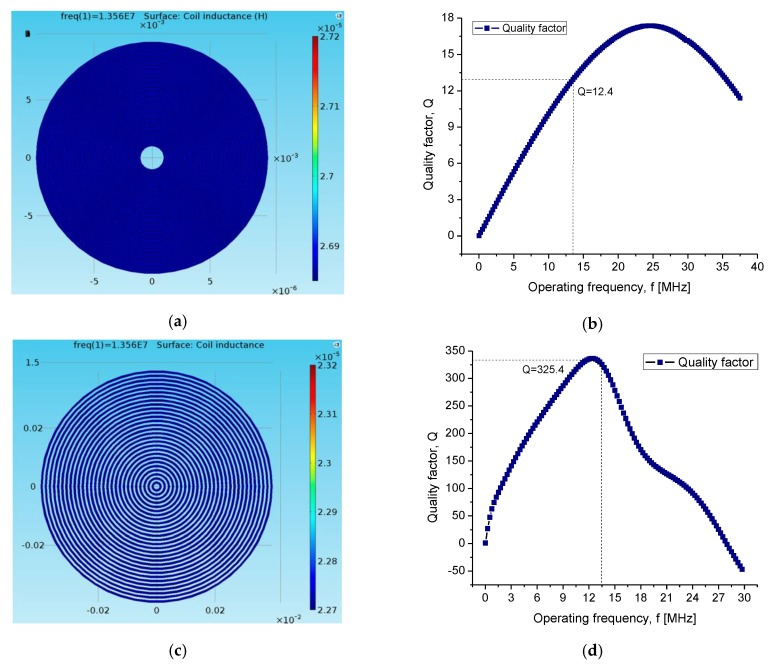
(**a**) Inductance characteristic of the internal coil; L = 25.6 μH, R = 280.8 Ω, Q = 12.4. (**b**) Quality factor of the internal coil, (**c**) inductance of the external coil; L = 20.18 μH, R = 6.61 Ω, Q = 325.4. (**d**) Quality factor of the external coil.

**Figure 9 sensors-20-01525-f009:**
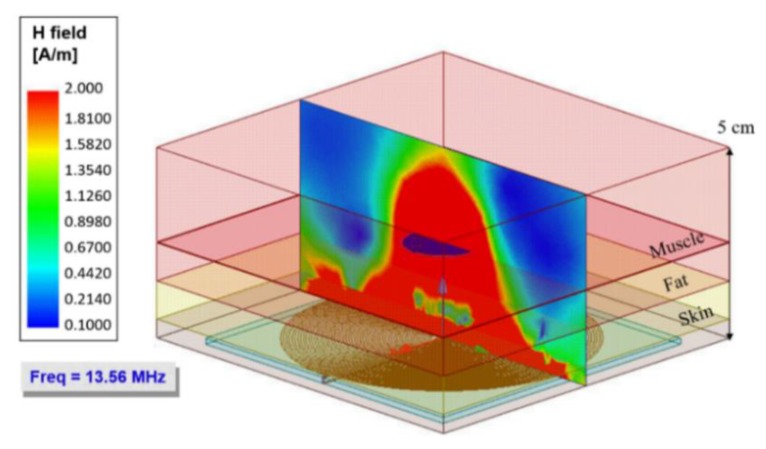
The intensity of the magnetic flux for the inductive power transfer (IPT) system at 13.56 MHz. The bottom plane represents the external coil (radiating source).

**Figure 10 sensors-20-01525-f010:**
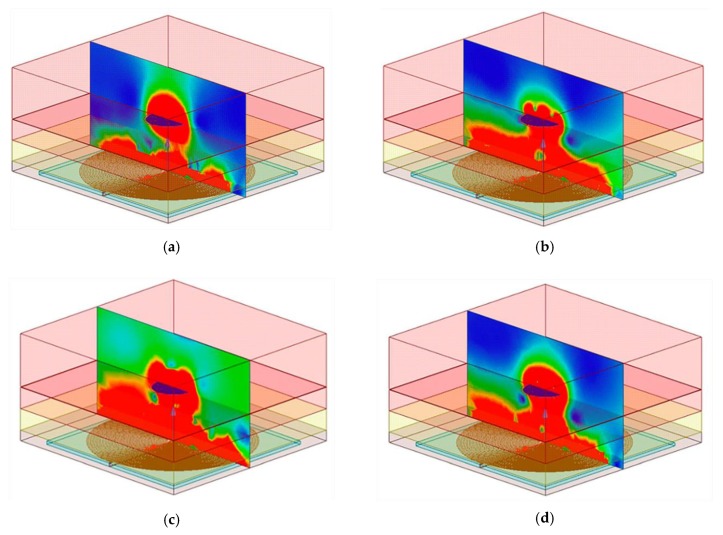
Variation of the magnetic flux density at (**a**) 4.5 MHz, (**b**) 7 MHz, (**c**) 9.5 MHz, and (**d**) 12 MHz.

**Figure 11 sensors-20-01525-f011:**
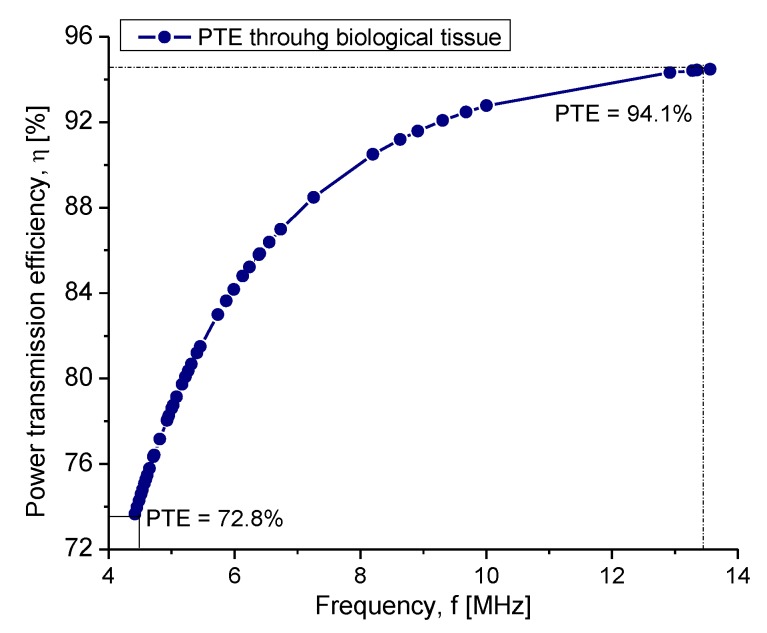
Power transmission efficiency according to the operating frequency system.

**Figure 12 sensors-20-01525-f012:**
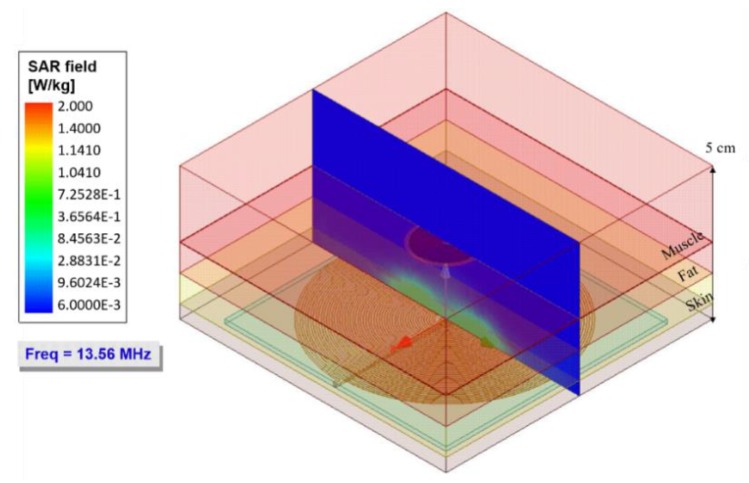
The specific absorption rate for the bidirectional IPT system at the operating frequency of 13.56 MHz.

**Figure 13 sensors-20-01525-f013:**
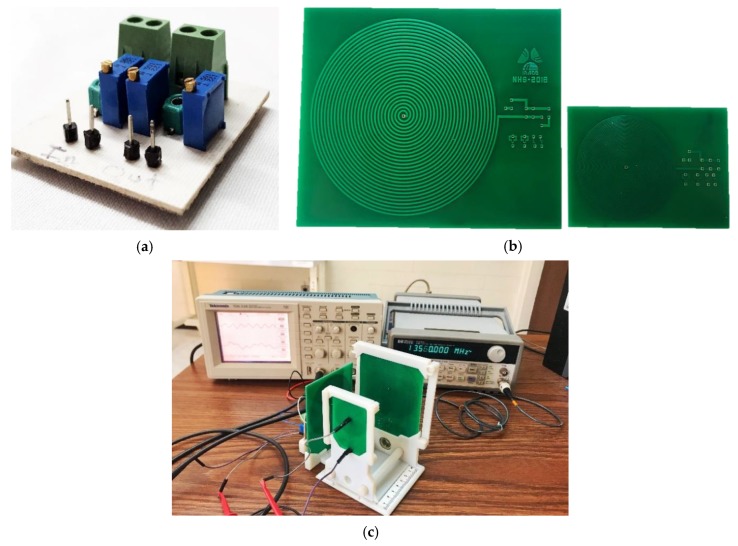
Modules manufactured on PCB-FR4 substrate. (**a**) Bridge circuit and (**b**) internal and external coils. (**c**) Configuration for measuring the coupling at 3.5 cm distance between the coils.

**Figure 14 sensors-20-01525-f014:**
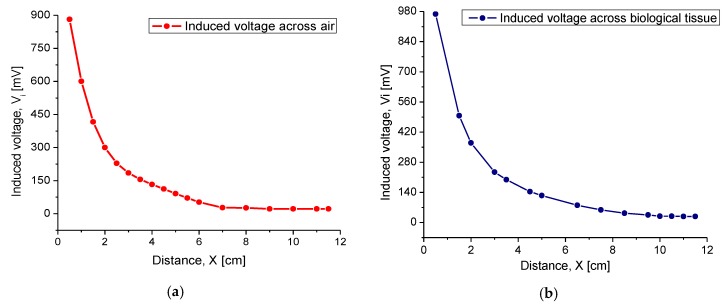
Induced voltage as a function of the separation distance between the coils for (**a**) air and (**b**) synthetic tissue.

**Figure 15 sensors-20-01525-f015:**
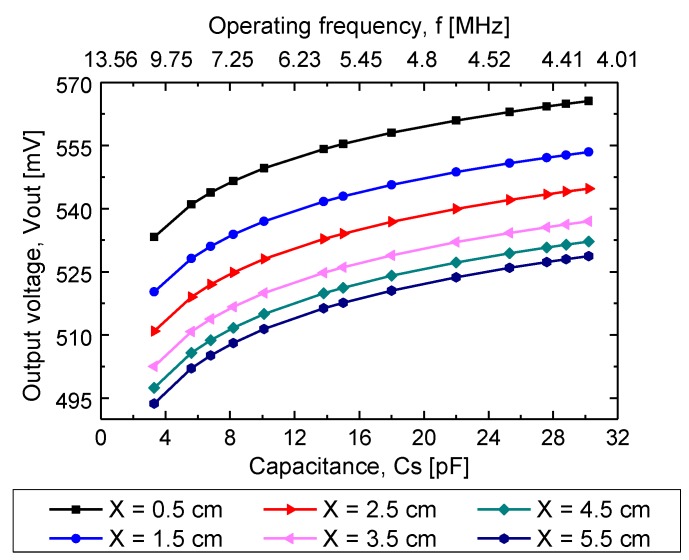
Experimental output voltage $V_{\mathrm{out}}$ as a function of capacitance and frequency at the implantable set, using a synthetic tissue as the magnetic core.

**Figure 16 sensors-20-01525-f016:**
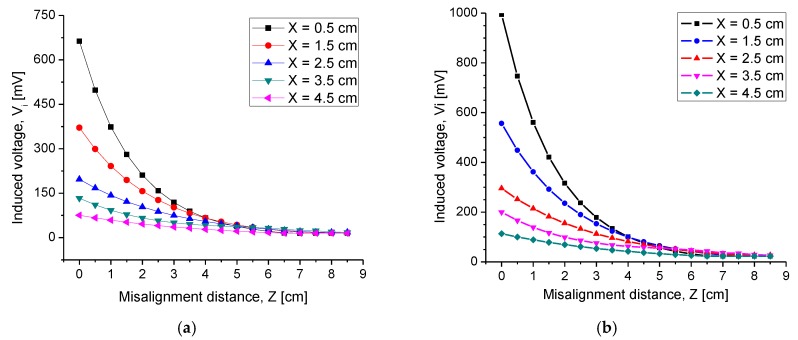
Characterization of induced voltage under transverse misalignment for different radiation distances through (**a**) air and (**b**) synthetic tissue.

**Figure 17 sensors-20-01525-f017:**
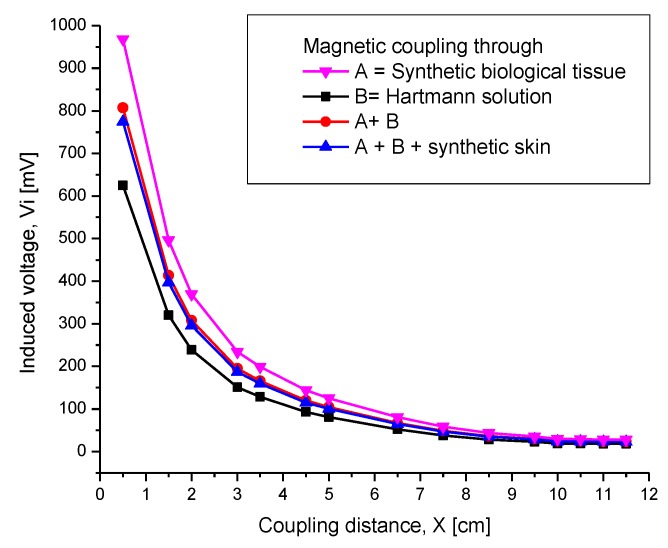
Induced voltage as a function of the separation distance between the coils combining different materials as the electromagnetic core.

**Table 1 sensors-20-01525-t001:** Constitutive parameters of human biological tissue at a frequency of 13.56 MHz [17].

Model	Thickness [cm]	Conductivity [Sm^−1^]	Relative Permittivity	Wavelength [m]
Skin	0.5	0.38421	177.13	2.87
Fat	1	0.030354	11.827	11.11
Muscle	2	0.62818	138.44	3.24

**Table 2 sensors-20-01525-t002:** Analytical results for the designed internal and external coils.

Quantity	Symbol	Internal Coil	External Coil
Internal diameter	Din	2 mm	2 mm
External diameter	Dout	18 mm	8 cm
Width	w	160 μm	700 μm
Thickness	h	1.5 μm and 1 μm	35 μm
Number of turns	N	28 each loop	27
length	l	1.14 m	1.77 m
Self-inductance	L	20.98 μH	21.24 μH
Electrical resistance	R	171.86 Ω	5.6 Ω
Quality factor	Q	11.5	354
Operating frequency	f_s_	13.56 MHz	
Radiation distance	X	3.5 cm	
Coupling coefficient	k	0.16	
Mutual inductance	M	3.79 μH	
Power transmission efficiency	η	92.%–74.5%	

**Table 3 sensors-20-01525-t003:** Printed circuit board (PCB) parameters for the fabricated internal coil.

Parameter	Symbol	Magnitude
Internal diameter	Din	2 mm
External diameter	Dout	3.8 cm
Width	w	250 µm
Separation	S	250 µm
Thickness	h	35 µm
Number turns	N	37
Length	l	1.3 m
Self-inductance	L	21.29 µH
Resistance	R	6.9 Ω

**Table 4 sensors-20-01525-t004:** Values of the bridge circuit components.

	Component	Magnitude	Units
Resistances	$R_{r}{,\;R}_{1}\;$	4.5	Ω
	$R_{2}{,\;R}_{3}$	47	Ω
Capacitances	$C_{1}$	6.8	pF
	$C_{r}$	50	pF
Inductances	$L_{1},{\;L}_{r}$	21.29	uH
	$L_{s}$	22.5	uH

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
