# Peer review of "Design of a Fully Integrated Inductive Coupling System: A Discrete Approach Towards Sensing Ventricular Pressure"

_sensors, 2020, doi:10.3390/s20051525_

Round 1

Reviewer 1 Report

Overall Comments:

The manuscript describes an interesting and well-designed method for creating a pressure sensor which might be implanted in the left ventricle. Although the contents of the manuscript are worthy of publication, there is much work that needs to be done on the structure and presentation of the results before the manuscript is of publishable quality. The Authors should restructure the manuscript to improve the narrative flow and highlight the important aspects of their study which they wish to present.

Major Comments:

The manuscript needs to be heavily rearranged to improve readability. Methods, results, discussion, and conclusion are all contaminated with elements that aren’t supposed to be there. The Author’s should either rearrange all elements into their appropriate sections or alternatively (and preferably) they may consider a non-traditional but more clearly labelled approach to presenting the results i.e 

Theoretical Separation Distance Design

Methods

Results

Surface Inductance

Methods

Results

Electromagnetic Flux Density

Methods

Results

Experimental Coupling Evaluation

Methods

Results

Etc.

The Author’s should then have a combined discussion, exploring all of the results from each section.

The conclusion should be much shorter than it currently is and state how well the objectives of the study were met and the clinical/scientific significance of the work. The Author’s should, in their discussion, compare their work to other published work in this field (of which there is a lot) – others outcomes, the Authors outcomes, and how the Author’s outcomes are an improvement. This will help to highlight the significance of this work. The Authors should include limitations and future work sections. For instance, the Authors designed a pressure sensor device but didn’t test changes in pressure at all! This will be a critical next step…

Minor Comments:

For reader convenience, all abbreviations should be defined in the abstract.

The introduction should discuss commercially available wireless implantable sensors for blood pressure monitoring. One highly successful example is the CARDIOMEMS which uses a similar wireless transmission principle but is obviously inferior to the Author’s concept as it is not continuous.

Where does the restriction for the 1.8 x 1.8 cm 2 sensor size come from? Is this for percutaneous implantation reasons (through the subclavian artery for example)?

Line 101 define “s”

Line 111 the Author’s should better define how k relates to (6)

Line 136 should read “it can be balanced”

Line 138-139 needs grammatical correction

Line 140 “Once the sensor set is implanted the distance between the coils remains constant…” Where is the sensor implanted? If it is inside the ventricle, then the distance between the coils may change as the patient changes position. It will also change slightly beat to beat. This would then change k and therefore give a variation in your final pressure reading.

Line 166 “? = (????  − ??? )/(? + ?)”   What is “w”? is “s” the same as previously defined?

Line 181 “?  is the number of turns and ? is the spacing between the metal lines” Is this the same s as before? It should be defined in the first instance. By metal lines do you mean the wire loops?

Line 187 “Table 1 shows the constitutive parameters of the biological tissue used as the magnetic core” Can the Authors reference these parameters?

The Authors might consider moving Figures 4, 6, and 7, and their accompanying text to the results section.

Lines 270-277 are all methods and should go in the methods section.

Lines 331-344 are all methods and should be in the methods section.

Figure 13a - Won’t building your bridge circuit on a protoboard incur large parasitic capacitive and inductive elements? Given you’re working with highly sensitive systems this might be significant and should be mentioned as a limitation.

Lines 389-390 “Characterization of possible magnetically decoupling between the coils was carried out by squeezing the transverse axes in distances of 0 cm to 8.5 cm” What increments were used?

Author Response

PAPER Journal Sensors 698607

REPLY TO REVIEWERS

REVIEWER #1

Reply to the major comments:

1 The manuscript needs to be heavily rearranged to improve readability. Methods, results, discussion, and conclusion are all contaminated with elements that aren’t supposed to be there.

            The manuscript was rearranged following the reviewer comments but also trying fitting for a scientific paper. Several paragraphs were rearranged and some other changed of section. A section 4.0  Discussion of results, was also included

            All the subsections were numerically ordered

            Table 1 was changed of section, for clarification purposes

            Figures 5 and 13a were modified, for a best technical description

  1. The conclusion should be much shorter than it currently is and state how well the objectives of the study were met and the clinical/scientific significance of the work.

            Conclusions section was shortened and highlighted the main results.

  1. The Authors should include limitations and future work sections. For instance, the Authors designed a pressure sensor device but didn’t test changes in pressure at all! This will be a critical next step…

            In the discussion section was stated that this paper is presenting the full design methodology, and while the system is under full fabrication, the prototype was tested using a PCB approach, the electromagnetic results considering the main parameters (including capacitance) are validating the proposed novel system. Of course the in vitro capacitance testing is to be a critical step but we have confidence in our thin film technology.

Reply to the minor comments:

  1. For reader convenience, all abbreviations should be defined in the abstract.

All abbreviations were defined in the abstract

  1. The introduction should discuss commercially available wireless implantable sensors for blood pressure monitoring. One highly successful example is the CARDIOMEMS which uses a similar wireless transmission principle but is obviously inferior to the Author’s concept as it is not continuous.

The following text including a new reference was added:

In 2006, began the development of a new class of implantable devices for aortic aneurysms and heart failure. The system was named CardioMEMS, and consist of an implantable pressure sensor, an external communication module and an intravenous system designed to deploy the sensor in the pulmonary artery. The battery-less 3.5 x 30 mm device has a wireless range of about 20 cm. However, it is not considered a continuous monitoring system, since to obtain a single measurement the patient must lie on a pillow, which is part of the external communication module and press a button.

  1. Where does the restriction for the 1.8 x 1.8 cm2 sensor size come from? Is this for percutaneous implantation reasons (through the subclavian artery for example)?

The size restriction comes from the implantation site. From the anatomical study of left ventricle, we determine that the specific site of implantation is in lower part of the ventricle, which has an approximate area of 1.8 x 18 cm2. See reference 16.

  1. Line 101 define “s”.

“s” is defined as a variable to simplify the analysis, it does not define a specific parameter, according to the article s = jw.

  1. Line 111 the Author’s should better define how k relates to (6).

The coupling factor k is directly related to the mutual inductance of equation (6).

Now this relation is defined in lines 135-136

  1. Line 136 should read “it can be balanced”.

Done. Now line 162

  1. Line 138-139 needs grammatical correction.

Grammar text was corrected

Now lines 164-165

  1. Line 140 “Once the sensor set is implanted the distance between the coils remains constant…” Where is the sensor implanted? If it is inside the ventricle, then the distance between the coils may change as the patient changes position. It will also change slightly beat to beat. This would then change k and therefore give a variation in your final pressure reading.

Yes, the sensor will be implanted in the left ventricle. Indeed, changes can occur in the separation distance between the coils altering the coupling factor. However, in our theoretical consideration we are stating that this distance remains constant. During simulation routines, it allows analyzing the influence of these inherent variations into the system. The final result is validated after the electromagnetic characterization.

  1. Line 166 “? = (???? − ??? )/(? + ?)”   What is “w”? is “s” the same as previously defined?.

w is the width of the turns of the coil. This definition was added to the article.

Now line 186

  1. Line 181 “? is the number of turns and ? is the spacing between the metal lines” Is this the same s as before? It should be defined in the first instance. By metal lines do you mean the wire loops?

N is the number of turns, s is effectively the spacing between the metal lines, it is not the same s as before. The previous s has already been defined just like this one. The metal lines are the wire loops.

  1. Line 187 “Table 1 shows the constitutive parameters of the biological tissue used as the magnetic core” Can the Authors reference these parameters?

Yes, the reference was added, [17].

  1. The Authors might consider moving Figures 4, 6, and 7, and their accompanying text to the results section.

This suggestion and some others were seriously considered, but finally and attending the central idea of the authors, we decided not move these figures.

  1. Lines 270-277 are all methods and should go in the methods section.

This paragraph was included in the section 3.0 named Electromagnetic Flux Simulation, trying to give clarity to the full article.

  1. Lines 331-344 are all methods and should be in the methods section.

These lines were removed.

  1. Figure 13a - Won’t building your bridge circuit on a protoboard incur large parasitic capacitive and inductive elements? Given you’re working with highly sensitive systems this might be significant and should be mentioned as a limitation.

At a second characterization stage, the bridge circuit was built on a printer circuit board using variable capacitor and resistors to facilitate the balancing of the circuit. The new corresponding image was included.

  1. Lines 389-390 “Characterization of possible magnetically decoupling between the coils was carried out by squeezing the transverse axes in distances of 0 cm to 8.5 cm” What increments were used?

The characterization of possible magnetically decoupling between the coils was carried out by squeezing the transverse axes in distance of 0 cm to 8.5 cm, in steps of 0.5 cm.

Reviewer 2 Report

Review of the paper with the title: “Design of a full-integrated inductive coupling system: A discrete approach towards sensing ventricular pressure” by Natiely Hernández Sebastián a.o.

The paper presents the results of an analysis and modeling of a sensor that is intended for monitoring of ventricular pressure. The work, although brings relatively little novelty to the field, presents the design concept, the modeling, the results of the simulation and some experimental results performed in laboratory. I believe that the authors realize how far away from implementation their design is.

As there is not a lot of research subject in here,

I will consider the paper as a design report. From the novelty point of view is very little that I could praise the work. I would guess that the work was carried out as a design project as the research is rather limited.

The stream of the presentation is correct, the results are correctly presented, they make sense and I would say, such paper is quite useful for those who need to measure some deformation remotely.

I would have no problem at all with seeing the paper published, although the write-up needs some review (I believe the editors may contribute to the quality improving of the write-up).

I wish to suggest the authors to include in the paper some results and discussion related to the inductive coupling through different materials and non-homogenous media,

Author Response

PAPER Journal Sensors 698607

REPLY TO REVIEWERS

Reviewer #2

  1. I wish to suggest the authors to include in the paper some results and discussion related to the inductive coupling through different materials and non-homogenous media.

Reply:

We have realized a new electromagnetic characterization set, mainly including Hartmann solution, looking for explore some highly attenuating media, the overall response was included in the section 4.2.4 for Combined Electromagnetic Core and figure 17.

Round 2

Reviewer 1 Report

Overall the manuscript has been improved greatly and is now at a quality acceptable for publishing.

There are several minor comments which should be addressed, mostly grammatical: 

112 I think this should say “according to anatomy room restriction”

179 “The bridge circuit it can be balanced” I think should be “The bridge circuit can be balanced”

328 copper misspelt as cooper

514-516 – I’m not sure I understand this sentence I think there might be a grammatical mistake in here somewhere.

517 Should keep tense consistent (is/ was)

583-584 I think there is a typo here should read something like “…decoupling between the coils was carried out by squeezing the transverse axis of the x in distances of y from 0 to 8.5 cm”

657-658 How close is “close” for “the results are very close with the Hartmann solution”

684-686 This line is noticeably copied from the results (589-590)

705 should read “According to the”

705-708 Sentence needs some grammatical correction.

The manuscript could definitely use some professional English editing. I would encourage the Authors to seek this out, or otherwise take great care in proof-reading the manuscript.

I recommend this manuscript for publication without further peer review pending amendment of the minor comments outlined above.

Author Response

All the grammar corrections have been done